# Application of Near Infrared Spectroscopy for the Rapid Assessment of Nutritional Quality of Different Strawberry Cultivars

**DOI:** 10.3390/foods12173253

**Published:** 2023-08-29

**Authors:** Manuela Mancini, Luca Mazzoni, Elena Leoni, Virginia Tonanni, Francesco Gagliardi, Rohullah Qaderi, Franco Capocasa, Giuseppe Toscano, Bruno Mezzetti

**Affiliations:** Department of Agricultural, Food and Environmental Sciences, Università Politecnica Delle Marche, Via Brecce Bianche 10, 60131 Ancona, Italy; m.mancini@pm.univpm.it (M.M.); l.mazzoni@staff.univpm.it (L.M.); e.leoni@staff.univpm.it (E.L.); virginiatonanni@gmail.com (V.T.); s1071951@studenti.univpm.it (F.G.); r.qaderi@pm.univpm.it (R.Q.); f.capocasa@staff.univpm.it (F.C.); g.toscano@staff.univpm.it (G.T.)

**Keywords:** phenotyping, breeding, classification, PLS-DA, data fusion, non-destructive

## Abstract

Strawberry is the most cultivated berry fruit globally and it is really appreciated by consumers because of its characteristics, mainly bioactive compounds with antioxidant properties. During the breeding process, it is important to assess the quality characteristics of the fruits for a better selection of the material, but the conventional approaches involve long and destructive lab techniques. Near infrared spectroscopy (NIR) could be considered a valid alternative for speeding up the breeding process and is not destructive. In this study, a total of 216 strawberry fruits belonging to four different cultivars have been collected and analyzed with conventional lab analysis and NIR spectroscopy. In detail, soluble solid content, acidity, vitamin C, anthocyanin, and phenolic acid have been determined. Partial least squares discriminant analysis (PLS-DA) models have been developed to classify strawberry fruits belonging to the four genotypes according to their quality and nutritional properties. NIR spectroscopy could be considered a valid non-destructive phenotyping method for monitoring the nutritional parameters of the fruit and ensuring the fruit quality, speeding up the breeding program.

## 1. Introduction

Strawberries are one of the most widely cultivated berry fruit globally, with an annual production of 13.3 million tons in an area spanning 522,527 hectares [1]. The expectations of consumers regarding their health are closely tied to the inherent characteristics of strawberries, particularly the presence of bioactive compounds that have a positive impact on overall wellbeing. Numerous epidemiological studies have consistently demonstrated that a diet rich in fruits and vegetables is often associated with a reduced risk of various chronic ailments, including obesity, infections, cancer, and cardiovascular and neurological diseases [2,3]. In the realm of fruits, berries, including strawberries, play a significant role due to their high phytochemical content [4,5,6]. Extensive scientific research has confirmed that strawberries contain bioactive substances with potent antioxidant properties, such as ascorbic acid, polyphenolic compounds (ellagic acid and ferulic acid), as well as various flavonoids (such as anthocyanins, catechins, and phenolic acids). These compounds have nutraceutical effects, meaning they offer beneficial and protective properties for the human body [7,8,9].

Conventional approaches for assessing the quality characteristics of fruits typically involve destructive methods. While these methods provide clear and reliable information, they have limitations in terms of analyzing many products due to the time and cost involved. Destructive techniques are generally applied to a representative sample rather than the entire production batch. Numerous studies have described destructive methods for measuring these parameters. For instance, sweetness is commonly evaluated using a manual or automatic refractometer, which determines the soluble solids content (SSC) expressed in Brix units [7,10]. The measurement of bioactive compounds like vitamins and polyphenols often involves techniques such as mass spectrometry, gas chromatography, and liquid chromatography [11]. In recent years, there has been growing interest among researchers in non-destructive and rapid analytical technologies, particularly those based on optical properties [11,12,13]. These alternative approaches aim to overcome the limitations of destructive methods and enable real-time evaluation and analysis of a broader range of samples, requiring minimal sample preparation [14]. Nondestructive methods usually take less time, are not harmful (no need for solvents or reactants) and have the additional benefit of not changing the fruit qualities during measurement. The immediate advantage is that such methods may be used to grade each fruit and vegetable according to their quality before being sold. Grading individual items is crucial for satisfying customer expectations since fruit and vegetable quality parameters exhibit a high degree of inherent variability. Since ancient times, color and size grading by visual examination has been used to both exclude items that would not satisfy the minimum standards for quality and to increase consistency. Over time, this assessment has been mechanized, and producers, cooperatives, and packing facilities all around the world are now utilizing high-speed grading lines that use sensors for exterior quality parameters including color, size, and appearance. New and intriguing marketing opportunities for horticultural goods have emerged with the development of nondestructive ways to quantify internal quality qualities, such as textural, nutritional properties or flavor, provided, of course, that the traits they measure are comparable to their human analogs. To improve post-harvest procedures, nondestructive methods are also highly helpful for creating models of how quality characteristics change during post-harvest storage. The inter-fruit variability and the time impact may be clearly distinguished since the same fruit can be watched throughout time. This significantly enhances the estimate of the kinetic parameters [15].

In recent years, several optical methodologies have succeed in assessing quality parameters like firmness, acidity, and soluble solids [16,17,18,19,20]. One particularly valuable alternative is near-infrared spectroscopy (NIR), which utilizes the infrared region of the electromagnetic spectrum to non-destructively examine the chemical and physical properties of samples. Spectroscopic techniques generate datasets containing vast amounts of data and information, which can effectively be analyzed using multivariate data analysis techniques or chemometrics. In particular, chemometrics help in managing common spectroscopic effects like peak shifts, scattering, interfering signals and baseline shifts [21]. The combination (fusion) of outputs from different instrumental techniques has gained great interest since it has the potential to increase the performance of categorization or prediction of food specifications in comparison to utilizing a single analytical approach. Although encouraging findings have been found in the authenticity and quality evaluation of food and drink, combining data from various methodologies is not simple and is an important task for chemometricians [22].

Recent studies have been conducted to assess the sensory and nutritional quality of strawberries using near-infrared spectroscopy. Our research group has a good experience on the use of NIR spectroscopy for the evaluation of strawberry fruits quality: in [16], NIR spectroscopy has been used to evaluate the possibility of developing good PLS models for the prediction of SSC, titratable acidity, firmness, and color. Good results were obtained only for firmness and SSC parameters, while the models obtained for the prediction of color and titratable acidity were not acceptable for screening quality control. In [20], sound and infected (by *B. cinerea* spores) strawberry fruits were analyzed by NIR spectroscopy. The aim of the study was to demonstrate that there was a high correlation between SSC and *B. cinerea* susceptibility and investigate the possibility of early detection of *B. cinerea*. Furthermore, the developed PLS regression model for the prediction of SSC content also confirmed the results of our previous study. NIR spectroscopy has also been used by other authors for predicting internal quality and discriminating among strawberry fruits from different production systems [23]. Instead, Andersen et al. used Raman spectroscopy to assess the chemical composition of strawberries. The total soluble solid content, fructose, glucose, sum of sugars, citric acid and sum of acids was predicted [24]. Other authors have evaluated the possibility of using NIR and ultraviolet-visible (UV-VIS) spectroscopy for predicting the number of days under storage of strawberry samples and as consequence their shelf-life [25]. The quality characteristics and consumer acceptability of strawberries were also evaluated by [26]. Quality depends on the ripening stages of the strawberry and pelargodinin-3-glucoside is the main anthocyanin that changes during postharvest distribution. Hyperspectral imaging technology (HSI), in the range of VIS-NIR and shortwave infrared (SWIR) light, was used to assess its content according to two types of harvest maturity. However, these studies typically suffer from limitations such as a small number of selected fruits, which can diminish the accuracy of the predictive models developed [27]. Furthermore, only a few strawberry cultivars are usually tested, sometimes even just one, and they often involve different sample treatments (e.g., varying pesticide treatments, cultivation systems, and post-harvest storage conditions). Additionally, multiple quality parameters are simultaneously considered [23,28,29]. It is to be noted that the mentioned studies have used regression algorithms for the prediction of the parameters of interest of strawberry samples.

To the best of our knowledge, the use of classification algorithms in the literature is mainly used for qualitative characteristics but it was never used as a phenotyping tool for the rapid assessment of the global nutritional and qualitative aspects of strawberry cultivars. The great novelty of this study is related to the application of NIR as a non-destructive tool for the evaluation of strawberry nutritional quality, and the use of data fusion techniques to determine the feasibility of developing good classification model for assessing the quality of strawberry fruits. To this aim, important qualitative characteristics of four strawberry genotypes, including soluble solids, vitamin C, anthocyanins, and phenolic acids content, have been assessed by using traditional destructive methods. Partial least squares discriminant analysis (PLS-DA) has been used to develop a model that could classify the four selected genotypes according to the quality and nutritional parameters. The introduction of non-destructive phenotyping methods enables the monitoring of quality parameters, ensures fruit quality prior to sale, and facilitates the development of predictive models to expedite the selection process.

## 2. Materials and Methods

### 2.1. Plant Material and Experimental Plan

The assessment of qualitative parameters was conducted on four different strawberry cultivars: “Cristina”, “Romina”, “Sibilla”, and “Silvia”. These strawberries were grown in the 2019–2020 season at the Didactic and Experimental Farm Center for Agricultural Research “P. Rosati” in Agugliano, Italy (coordinates: 43°320′ N–13°220′ E). The cultivation method followed the typical practices of the Marche Region. Cold-stored “A” type plants were used, and they were planted in open fields at the end of July. Each genotype consisted of a total of 32 plants. A selection of 27 fruits was made from each genotype, which were then sent to the laboratory for analysis. The fruits were collected in two separate harvests, resulting in a total of 54 fruits per genotype. Non-destructive NIR spectroscopy was used to analyze each fresh fruit individually. Subsequently, the fruits were frozen individually and analyzed after a few days to determine their soluble solids content, titratable acidity, vitamin C content, anthocyanin content, and phenolic acids content.

### 2.2. Soluble Solids Content Analysis

A digital refractometer (Palette PR101, Atago, Tokyo, Japan) with automatic temperature compensation was used to measure the soluble solids content (SSC) of each fruit. The frozen fruits were individually thawed before this analysis, which had been conducted after they had been examined by NIR. A few drops of the juice were then applied to the refractometer slide after they were squeezed to remove the juice. Each fruit °Brix value was measured twice [16].

### 2.3. Titratable Acidity Analysis

The automatic titrator HI 84532 Fruit Juice Titratable Acidity (Hanna Instruments, Woonsocket, RI, USA) was used to quantify the titratable acidity (TA). A plastic beaker was filled with 5 mL of the previously acquired juice and 45 mL of ultrapure water. The manufacturer-supplied titrating solution was used by the equipment to automatically titrate this solution until pH 8 was reached. Citric acid percentage (% citric acid) is used to measure acidity [30].

### 2.4. Methanolic Extraction

The fruit extracts were prepared following the method described by Diamanti et al. [31]. Initially, 2 g of fruits were homogenized using an Ultraturrax T25 homogenizer (Janke and Kunkel, IKA Labortechnik, Staufen, Denmark) in 4 mL of methanol. The homogenization process was carried out for 30 min in a dark environment. Afterward, the suspension was subjected to centrifugation at 4500 rpm for 10 min at a temperature of 4 °C. The resulting supernatant was collected, while the pellet containing the fruit remnants was subjected to a second extraction. For the second extraction, another 4 mL of methanol was added to the pellet, and the procedure was repeated. The second supernatant obtained was combined with the first supernatant. The combined supernatants were then immediately injected into the High-Performance Liquid Chromatography (HPLC) system for further analysis.

### 2.5. Vitamin C Extraction

To extract Vitamin C from the fruits, an ultrasound-assisted extraction method was employed, following the protocol described by Tulipani et al. [32]. The extraction process involved the use of an ultrasound bath (Bioblock/ELMA 88155, Stuttgart, Germany), which generates ultrasound waves within a water-filled tank using high-frequency electric current provided by a generator. This method enhances the dissolution of solutes in specific solvents, thereby expediting the extraction process. For the analysis, 1 g of frozen strawberries was homogenized with a 4 mL portion of the extraction buffer solution, which contained 5% metaphosphoric acid and 1 mM DTPA. The homogenization was carried out for 5 min using sonication. Subsequently, the mixture underwent centrifugation at 4000 rpm for 10 min at a temperature of 4 °C. The resulting supernatants from each sample were then filtered using a 0.45 μm pore size filter and transferred to a vial for further analysis on an HPLC system.

### 2.6. Determination of Vitamin C Content

The method by Helsper et al. [33] outlined for measuring vitamin C content was used. The extraction process was followed by HPLC examination of the extracts. The HPLC system included a Jasco PU-2089 plus controller, a Jasco UV-2070 plus ultraviolet (UV) detector, set at an absorbance of 260 nm, and a Jasco AS-4050 autosampler, all from Jasco in Easton, Maryland, in the United States. A Phenomenex 4.0 × 3.0 mm C18 ODS guard column (Phenomenex, Torrance, CA, USA) was utilized to protect the Ascentis Express C18 150 × 4.6 mm HPLC column (Supelco, Bellefonte, PA, USA). The gradient program used two mobile phases, A (50 mM phosphate buffer, pH 3.2) and B (acetonitrile), with A present at 100% for the first 6 min, 50% for the next 2 min, and 100% for the remaining time. The amount of vitamin C in strawberries was measured using a calibration curve created by executing standard vitamin C concentrations. The results were represented as mg of vitamin C per 100 g of fresh weight (FW).

### 2.7. Determination of Phenolic Acid Content

In accordance with the procedures outlined by Schieber et al. [34] and Fredericks et al. [35], the analysis of phenolic acids was carried out. A Jasco PU-2089 plus controller, a Jasco UV-2070 plus ultraviolet (UV) detector, and a Jasco AS-4050 autosampler made up the HPLC system used for the analysis. Jasco is based in Easton, Maryland. The column used was an Aqua Luna C18 250 4.6 mm (Phenomenex, Torrance, CA, USA) protected by a Phenomenex 4.0 3.0 mm C18 ODS guard column (Phenomenex, Torrance, CA, USA). The HPLC UV detector was set at a wavelength of 320 nm. Two mobile phases made up the gradient program: A (2% acetic acid) and B (1:50:49 acetic acid, acetonitrile, and water). The gradient program began with 55% A and 45% B for 50 min, then 100% B for 10 min, and finally a decline to 10% B until the completion of the run. Calibration curves were built using external standards of chlorogenic acid, caffeic acid, and ellagic acid for the quantification of phenolic acid concentration. The results were reported in milligrams (mg) of phenolic acids per 100 g of fresh-weight strawberries (mg/100 g FW).

### 2.8. Determination of Anthocyanin Content

The method described by Fredericks et al. [35] was used to analyze the anthocyanin content. A Jasco PU-2089 plus controller, a Jasco UV-2070 plus ultraviolet (UV) detector, and a Jasco AS-4050 autosampler made up the HPLC system utilized for the analysis. Jasco is based in Easton, Maryland, in the United States. The chemicals were separated on a Phenomenex Aqua Luna C18 (2) reverse-phase column (250 4.6 mm) with 5 µm particle size. For column protection, a Phenomenex 4.0 mm × 3.0 mm C18 ODS guard column was used. A wavelength of 520 nm was used to monitor the chemicals. Two mobile phases—A (formic acid, acetonitrile, and water in a ratio of 10:3:87) and B (formic acid, acetonitrile, and water in a ratio of 10:50:40)—made up the gradient program. The gradient program started at 75% A for 10 min, dropped to 69% A for 5 min, then 60% A for another 5 min, and finally resumed at 50% A for 10 min. Finally, 90% A for 16 min brought the session to a close. Calibration curves for the determination of anthocyanin content were created using cyanidin-3-glucoside, pelargonidin-3-glucoside, and pelargonidin-3-rutinoside as external standards. The results were expressed as milligrams (mg) per 100 g of fresh strawberries.

### 2.9. NIR Analysis

In this work, the analysis was carried out using a Fourier Transform (FT) NIR spectrophotometer (FT-NIR mod. Nicolet iS10, Thermo Scientific™, Waltham, MA, USA), equipped with an integrative sphere (Smart NIR Integrating Sphere, Thermo Scientific™, Waltham, MA, USA). The spectra were collected in the 10,000 to 4000 cm^−1^ region of the near-infrared spectrum. To minimize the moisture content in the instrument throughout the assessment and lower the variability of the spectral analysis, the analysis was conducted utilizing a constant flow of nitrogen. Each spectrum was captured with an average of 32 scans at 4 cm^−1^ resolution, yielding 1557 absorbance data. To reflect the whole electromagnetic signal and minimize variations caused by the environment rather than the sample, the background spectrum was examined hourly.

As soon as the fruit was harvested and before it was frozen, it was examined twice. The fruit was spun 180 degrees for the second measurement after the initial measurement was made at a position along its equator. The final dataset consists of 216 samples (432 observations × 1557 wavenumbers).

### 2.10. ANOVA

To assess whether there were statistically significant differences among the cultivars for the various qualitative parameters (SSC, TA, vitamin C, anthocyanins, and phenolic acids content), one-way analysis of variance (ANOVA) was conducted at a confidence level of 95%. After performing the ANOVA, Tukey’s post hoc test was applied to determine significant differences among the groups. This post hoc test compares all possible pairs of means to identify which groups differ from each other. Differences with a *p*-value less than 0.05 were considered statistically significant. The post hoc test is based on the studentized range distribution and is commonly used for multiple pairwise comparisons following an ANOVA analysis.

### 2.11. Multivariate Data Analysis

As a first step, principal component analysis (PCA) was used as explorative method for reducing the dimensionality of the dataset and investigating correlations between variables and similarities between samples [36]. PCA was computed separately on the two different datasets, i.e., the lab analysis data (D-Lab) with size 432 × 5 and the NIR spectral data (D-NIR) with size 432 × 1557. Before model computation, missing data were removed from D-Lab dataset and consequently some entries/samples were also deleted from D-NIR dataset. The resulting dataset consists of 394 observations × 1557 wavenumbers (197 samples). The presence of missing data is due to physical and experimental issues that happened during the fruit analyses (low amount of fruit for the analysis of all the parameters, problems during the extraction/analyses of samples in the laboratory, the impossibility of repeating the missing analyses in this single-fruit study).

In order to eliminate unwanted physical phenomena from the D-NIR dataset, several preprocessing techniques were used, including Standard Normal Variate (SNV), Multiplicative Scatter Correction (MSC), first and second derivative spectra (Savitzky-Golay filter [37] with 9 or 13 smoothing points window and 2nd order polynomial degree), and a combination of the previous ones [21]. Finally, before any analysis, the spectra were always mean centered. Additionally, only the range between 9000 and 4000 cm^−1^ was included in the analysis to weed out variables that had too much noise and did not provide useful data. Prior to data analysis, the D-Lab dataset’s raw data were solely autoscaled. For each cultivar, confidence ellipses that used the mean score values as the center and the standard deviation of each variability direction as the radius were also computed to provide a clearer picture. Finally, loadings were examined to find the substances connected to sample separation in the PCA space.

Both low-level and mid-level data fusion approaches were adopted for the joint analysis of strawberry samples using the NIR data and lab analysis information. In the low-level strategy the fusion consists of concatenating column-wise the pretreated datasets and then analyzing the resulting dataset as one, having as many rows as the samples analyzed and several columns as the sum of the spectral wavenumbers and D-Lab columns. This new dataset consisted of variables with different measuring scales, so in order to compensate the variability of the different analytical techniques it was additionally normalized [22]. Block-scaling was used to prevent one block from being dominant over the other (data blocks are dimensionally unbalanced). It works by equalizing variance, so each block has variance equal to one, but it preserves the variance between variables inside each block. At last, mean centering was applied. Instead in the mid-level strategy, fusion occurs by concatenating features extracted from the different blocks. In this study the fused dataset (D-fused) was assembled by using five PCA scores from D-Lab and five PCA scores from D-NIR. To avoid different magnitudes between data blocks, autoscaling was used as data preprocessing for the fused data set.

Then, classification models were calculated both on the spectral and fused data sets. Partial least squares-discriminant analysis (PLS-DA) was used as a classification technique. It is based on partial least squares (PLS) algorithm and works by evaluating the relationship between a dummy matrix (or vector) reporting the class membership information as dependent Y block and an independent (spectral) matrix (X) [38,39]. In our study, strawberry fruits were divided into different categories based on the results (similarities/dissimilarities) of the PCA study and codification was 1/0 (belonging/not belonging to the category). Utilizing venetian blind-cross validation (5 segments) and an outside test set, all classification models were verified. Using the duplex approach, the dataset was divided into training and test sets [40]. For both the calibration and validation sets, our technique ensures a representative spanning of the entire data variability. We further verified by performing an exploratory data analysis that both sets fully cover the variability domain.

The ROC (receiver operating characteristic) curve was employed as an evaluation tool to assess the effectiveness of the generated model and its capacity to categorize unclassified samples. The ROC curve, which measures how well a model can distinguish between classes, is a probability curve. The true positive rate (TPR) and false positive rate (FPR) are represented on the *y*-axis of the ROC curve, which is a graph. The true positive rate (TPR) and False Positive Rate (FPR) are represented on the *y*-axis and *x*-axis, respectively, of the ROC curve.

It shows the performance of a classification model at all classification thresholds and defying a threshold limit on the *y* variable when a binary classification system is used to drive the decision. An excellent model has ROC near to the 1 which means good ability to recognize the samples belonging to the class and to reject the samples not belonging to the class.

Matlab (ver. R2022a, The MathWorks, Natick, MA, USA) and in-house functions based on existing algorithms were used for all data analysis.

## 3. Results and Discussion

This section is articulated in four parts. In the first one, the univariate ANOVA results are reported. In the second part a description of exploratory analysis results is presented for the separate datasets, namely, D-Lab and D-NIR datasets. In the third part, the fused dataset is considered, and the application of the mid-level approach is described in detail. In the last part the related classification models are reported.

### 3.1. ANOVA Results

The evaluation of SSC and TA revealed significant differences among the studied genotypes. For the first parameter, the cultivar “Sibilla” showed the highest SSC in fruits, with a value of 7.66 °Brix, followed by “Cristina” with 7.38 °Brix, which appeared to be significantly similar (Figure 1a). Then, “Romina” with 6.15 °Brix and “Silvia” with 4.83 °Brix which showed a significantly lower fruit sugar content. Regarding fruit TA, the higher value was registered again by “Sibilla” with 0.75%, like “Cristina” (0.68%). Also in this case, the lower values were registered by fruits of “Silvia” (0.65%) and “Romina” (0.48%), which resulted significantly lower than all the other cultivars (Figure 1b).

#### 3.1.1. Vitamin C

HPLC analysis of fruit vitamin C content divided the genotypes in two main groups, according to the amount of this compound: “Romina” and “Sibilla” showed similar and significantly higher content of vitamin C (31.84 and 31.33 mg/100 g, respectively) than “Silvia” and “Cristina” (15.64 and 14.31 mg/100 g, respectively) (Figure 2).

#### 3.1.2. Anthocyanin and Phenolic Acids

The main compounds belonging to the class of polyphenols that were analyzed in this study were anthocyanin and the phenolic acids. The HPLC analysis highlighted different trends for the two classes of compounds. Regarding anthocyanins, “Romina” significantly showed the highest fruit content with 93.81 mg/100 g, followed by “Silvia” (61 mg/100 g), “Sibilla” (48.98 mg/100 g), and “Cristina” (38.98 mg/100 g), which showed a similar result (Figure 3). For the phenolic acids content, fruits of “Sibilla” and “Silvia” showed significantly higher content (31.35 mg/100 g and 29.17 mg/100 g, respectively), while “Romina” and “Cristina” fruits registered similar values (21.03 mg/100 g and 19.16 mg/100 g, respectively), but significantly lower than the other two cultivars (Figure 3).

### 3.2. Spectra

Figure 4a reports the average raw spectra of the four cultivars considered in this study. In general, the absorbance spectra of the four cultivars were relatively similar and featureless, except for three major and broad absorption bands highlighted with dotted lines in the figure. The most deviating cultivar seems to be “Silvia”, while “Romina” was the cultivar with highest peak at 6860 cm^−1^ and “Sibilla” was the cultivar with lowest peak at 5188 cm^−1^. It is important to take into consideration that water constitutes about 80–90% of fruit and vegetables; consequently, the influence of water absorbance is very high [41] with broad bands having centers at approximately 970, 1200, 1450, 1950, and 2250 nm [42]. The second major component is carbohydrates. In general, NIR absorption bands are relatively broad and overlapping, and the assignment of the raw spectra is usually complex, beyond noting features related to water. What allows the use of NIRS over infrared spectroscopy is the development of chemometrics, which permitted the extraction of relevant data out of the spectra [43].

In fact, peak resolution could significantly improve with the use of spectral pretreatment. For this reason, spectra have been pretreated using a first derivative (Savitzky–Golay filter with 9 smoothing points window and second order polynomial degree) to reduce random noise (Figure 4b). Even if the absorption peaks are sharper and the pretreatment allows to better highlight the spectral differences, the interpretation of first derivative spectra is more difficult than of the raw spectra. In fact, a peak of maximum absorbance on the original spectra corresponds to zero in the 1^st^ derivative [44], as a consequence zero points corresponding to peak in the raw spectra were selected in Figure 4b. The band at 5188 cm^−1^ (3) is related to the OH bonds and is higher for Cristina and Silvia cultivars than for Sibilla and Romina [42,45]. The band at 6860 cm^−1^ (2) is higher for Silvia and Romina and is assigned to O-H and C-H combinations, related to qualitative parameters, such as the soluble solids content or the titratable acidity [45]. Lastly, the peak at 8580 cm^−1^ (1) is in the region of the second overtone of CH.

### 3.3. Principal Component Analysis

PCA was computed both on D-NIR and D-lab datasets separately. D-NIR data were pretreated using a first derivative (Savitzky–Golay filter with nine smoothing points window and second order polynomial degree) to reduce random noise. The first 3 PCs account for 93.26% of the total variance and the results are reported in Figure 5. It is not possible to see a clear separation among the four cultivars. A partial trend of separation was only observed between “Cristina” and “Silvia” (negative PC2 scores) vs. “Romina” and “Sibilla” (positive PC2 scores) on the scores plot of the PC1, PC2, and PC3 (data not shown). To determine the most significant wavenumbers responsible for this separation trend, the associated loading plot was examined. The band at 6850 cm^−1^ is related to the OH combination of water [41,45], as was previously noted in our earlier study [20], while the band at 5180 cm^−1^ is connected to the overtone of CH and OH bonds [46]. Since fruit and vegetables include between 80 and 90 percent water and O-H water bands dominate the NIR spectrum [41], carbohydrates defined by CH bonds make up the second main component.

D-Lab data were autoscaled before PCA computation and the results are reported in Figure 6. All 5 PCs retain part of the total variance, in detail PC1 = 26.4%, PC2 = 22.3%, PC3 = 19.7%, PC4 = 18.4%, and PC5 = 13.3%. As for D-NIR dataset, the four cultivars slightly overlap, and a partial trend of separation can only be observed on scores plot of PC2 vs. PC3. “Sibilla” and “Romina” cultivars are in the positive part of PC2 while “Silvia” and “Cristina” in the negative part. This trend is clearer by looking at Figure 6c which reports the standard error ellipses for each cultivar. The results are in line with our findings, particularly regarding the vitamin C content (ANOVA results Section 3.1). By the investigation of the loadings, it can be observed that vitamin C, anthocyanin and phenolic acids are mainly related to PC2; phenolic acid and anthocyanin are also described by (positive or negative, respectively) PC3 and SSC; and acidity contents by PC1.

### 3.4. Low- and Mid-Level Data Fusion

Since the two datasets are of different magnitude (size of datasets) the mid-level approach was preferred for partially overcoming the possible predominance of one data source over the other. In any case (for sake of clarity) the results of the low-level approach based on simple concatenation and scaling of the variables of different nature are reported in Appendix A. As for D-NIR and D-LAB PCAs, the four strawberry cultivars partly overlap, and it is only possible to observe a partial trend of separation on PC2 and PC3 (Appendix A). PC2 mainly distinguishes “Sibilla” and “Romina” cultivars from “Silvia” and “Cristina” cultivars. Inspecting the corresponding loading plots (Appendix A) it can be observed that vitamin C, SSC, and acidity contents are the main responsible for such separation. The contributing spectral regions in the NIR loading plot (Appendix A) have been associated with the presence of strawberry sugars and OH bonds [41,46,47].

In the mid-level strategy, the first five scores values of PCA computed on D-NIR and D-Lab datasets have been merged into a unique new block of variables. This is a key step since the way in which the different variables are concatenated, normalized, and scaled can affect the results. The main parameter to consider is the method to be used for feature extraction and the related number of features to retain. The type of scaling to apply is less critical than in low-level data fusion, because of the data reduction [48].

We are aware that we are considering five PCs in a dataset of size 394 × 5 but the explained variance for each PCs from one to five is, respectively, 26.4, 22.3, 19.7, 18.4, and 13.3%. Therefore, we deemed not overestimating the model retaining all five PCs. For D-NIR PCA, the same number of PCs were selected and they cumulated 95.12% of the total variance. Selecting the same number of PCs for the two datasets, the respective data blocks had the same weight in the multi-block array, now suitable for further data processing. In the low-level strategy the loadings (P) of the model derive from the concatenated D-Lab and D-NIR datasets and, consequently, their interpretation is rather easy. They were computed through the PCA relation X = T * P^T^ + E where X was the concatenated datasets, T were the scores, and E the residuals. Instead, in the mid-level strategy the contribution of the original variables (lab analysis and spectral variables) in determining the final PCs was unknown and could be computed with the formula: O = P^T^ * F where F was the column-wise concatenation of the individual loadings obtained in the two separated PCA and P^T^ were the loadings obtained from PCA performed into the merged scores matrices (D-fused dataset). Transforming back the loading block-wise allowed us to investigate the relation of the loadings of the final PCA with the original variables. In this way, it was possible to understand which variables (i.e., lab analysis or spectral variables) were most involved in the separation of strawberry samples in the PCA space [49,50,51].

The results on fused data are presented in Figure 7. The fused and autoscaled data showed a clustering of the four cultivars similar to the one observed for D-Lab PCA and D-NIR PCA, as reported in Figure 5 and Figure 6. The PC1 vs. PC2 score plot separates the samples of “Sibilla” and “Romina” from “Silvia” and “Cristina” along the first principal component (Figure 7a). To our knowledge the former group has higher SSC and acidity contents with respect to the second (see Section 3.1). Much more interesting is the distribution of the samples in the score plot of PC2 vs. PC3 (Figure 7b). In fact, “Romina” and “Cristina” are in the bottom right part of the score plot (negative PC3 values and positive PC2 values), while “Silvia” and “Sibilla” are in the positive part of PC3. Thanks to the reconstruction of loadings of the D-Lab original variables, it is possible to visualize that this samples pattern is mainly attributed to quality and nutritional parameters (Figure 7a,b). As also confirmed by the ANOVA results, “Cristina” is the cultivar with lower nutritional parameters (vitamin C, anthocyanins, and phenolic acids), preceded by “Romina” characterized by good anthocyanin and vitamin C contents. “Silvia” and “Sibilla” have similar nutritional characteristics, with the former rich in vitamin C and phenolic acids and the latter characterized by higher SSC and acidity contents. Figure 7c reports the loadings plot of the three first PCs from the mid-level data fusion (D-fused).

### 3.5. Classification Models

Classification models were developed both on the spectral dataset (D-NIR) and on the D-fused dataset. Each sample was averaged across the two NIR replicates before computation. The two datasets were split into training and test sets of 137 and 60 samples, respectively. PLS-DA models have also been developed on the single D-NIR dataset to confirm the superior characterization achieved by data fusion observed by PCA results.

As shown by the PCA score plots (see Section 3.3 and Section 3.4), “Romina” and “Sibilla” cultivars have similar characteristics in all three different PCA approaches tested, mainly related to SSC and acidity content. “Cristina” and “Silvia” cultivars are on the opposite side of the PCA score plots, with “Silvia” characterized as the cultivar with the highest acidity content and “Cristina” the poorest in nutritional parameters. For these reasons, we have decided to develop three different PLS-DA models for classifying: (i) “Romina” and “Sibilla” from “Silvia” and “Cristina” cultivars (MODE 1); (ii) “Silvia” from the other cultivars (MODE 2) and (iii) “Cristina” from the other cultivars (MODE 3). Please note that we have also developed a multiclass PLS-DA model for classifying all cultivars simultaneously without obtaining any good results (the percentage of correctly classified samples is 37.0%, 55.9%, 53.9%, and 48.0% for “Romina”, “Sibilla”, “Silvia”, and “Cristina” cultivars, respectively). This is probably related to similarity in the spectral characteristics between some cultivars (e.g., “Romina” and “Sibilla”) making it difficult to develop a classification model able to recognize them. In any case, it is important to consider that the classification model will be used as phenotyping tool in the breeding process where it is helpful to have rapid information about qualitative and nutritional characteristics more than recognize one cultivar from all the another.

The results obtained, which are reported in Table 1, according to the classification criterion described in Section 2.11, show records for each dataset the data preprocessing, the model dimensionality (assessed by cross-validation), and classification performance (assessed by ROC curve). In detail, sensitivity (or true positive rate—TPR), specificity (true negative rate—TNR), and the misclassification error were used as a measure of the classification performance of the PLS-DA models. The results showed that classification models can discriminate the strawberries of “Cristina” and “Silvia” from “Sibilla” and “Romina” cultivars both using the fused dataset or only the spectral data. Sensitivity values (TPR) suggest that the model developed with the spectral data can correctly classify the samples (95.4% for training set and 100.0% for test set) and the specificity values (TNR) indicate that the same model can reject the samples of the other class (93.1% for training set and 100.0% for test set). With regard to the fused dataset, less LVs are needed in the development of MODE 1 model obtaining similar results. In detail, the misclassification error in cross-validation decreases from 5.8% to 4.4% while in validation it increases from 0.0% to 6.7%. This could be simply related to the dimensionality of the test set and to the fruit variability withing each cultivar. The latter can increase the error in classification as quality is more fruit specific than cultivar specific.

Both MODE 2 and MODE 3 developed on the D-NIR dataset have higher classification errors demonstrating that the models are not able to recognize only “Silvia” or “Cristina” cultivars, respectively. This is mainly due to the TNR values, so it is related to the ability of the models in correctly classifying “Silvia” or “Cristina” samples in the class. In fact, despite the high number of samples collected and analyzed, the dataset needs to be further increased especially for better covering all the variability domain of the different cultivars. For this study a total number of 51 and 49 strawberries have been collected for “Silvia” and “Cristina”, respectively. The possibility of combining the information from the different instruments by using data fusion strategy was also investigated for MODE 2 and MODE 3. The results are promising. The misclassification error is in general lower than 20% apart from the cross-validation phase of the PLS-DA model for the classification of “Silvia” cultivar (21.9%). More specifically, the error in validation of MODE 2 decreases from 26.7% to 13.3%, while for MODE 3 it decreases from 28.3% to 11.7%. So, the models could be used for quality control applications. In general, it is worth applying data fusion approaches because the features’ reduction step (for this study we have used PCA) removes part of the non-informative variance from the blocks. As a result, when data fusion is used for classification (or regression), often they provide better classification (or prediction) than using the separate datasets. These results are consistent with the fact that the global scores plot of D-fused dataset show a better separation among the different cultivars both according to acidity and SSC contents (PC1 vs. PC2) and to the nutritional parameters (PC2 vs. PC3).

Even if the D-fused showed an increased separation trend, it is important to consider that the advantage of performing the classification directly on the spectral data is the speed and cost of the analysis. In fact, near-infrared spectroscopy could be a valid solution for getting economic and fast information about the taste and nutritional quality, helping in speeding up the breeding process according to consumer behavior acceptance [52]. Before using D-fused dataset for modelling we have to consider how many lab measurements could be afforded/performed in addition to NIR to retrieve the cultivar information.

Figure 8 illustrates the scores scatter plot for the first two latent variables of the PLS-DA models obtained by using the spectral dataset (Figure 8a) and the D-fused dataset (Figure 8b) for the classification of strawberries samples of “Silvia” or “Cristina” cultivars from “Sibilla” or “Romina” cultivars. In comparison to the model developed on the D-NIR dataset, the PLS-DA model on D-fused dataset shows a separation trend between the two pair of cultivars (“Cristina” and “Silvia” vs. “Sibilla” and “Romina”) even if it is missing a clear separation. In detail, the two categories are more overlapping in the scores plot of Figure 8a,b. This is also confirmed by the better classification results of the latter model.

In Figure 9 the loadings plot, showing the contribution of the original wavenumbers to the final PLS-DA classification model of D-NIR dataset, is reported. The wavenumbers at 6930 and 5320 cm^−1^ are mainly related to the OH, CH, and CH_2_ deformation, while the band at 5195 cm^−1^ is assigned to OH combination bonds [46]. The selected wavenumbers match with the same bands selected by the recursive weighted partial least squares (rPLS) method for the prediction of SSC content of strawberry fruits in our previous study [20], highlighting the influence of this parameter for the separation of the different cultivars.

## 4. Conclusions

This study led to results of great interest in the field of strawberry fruit analysis, in particular the prediction of nutritional quality through non-destructive NIR spectroscopy technique. The main findings can be summarized as follows.

Fruit quality is strongly affected by genotypes: “Sibilla” and “Cristina” presented higher values of SSC and TA, but lower values of ACY; “Sibilla” and “Romina” presented higher values of vitamin C; “Sibilla” and “Silvia presented high values of phenolic acids, while “Romina” showed the highest values of ACY fruit content.PCA showed a trend of separation among cultivars based on their quality characteristics. In general, it can be stated that “Romina” is the cultivar with highest vitamin C and anthocyanin contents, “Sibilla” and “Silvia” the cultivars with highest phenolic acid content, and “Cristina” is the poorest in nutritional parameters.Classification models were developed based on spectral data (D-NIR) and a fused dataset (D-fused), i.e., combining D-NIR and D-Lab data using a mid-fusion strategy. The PLS-DA models successfully discriminated between “Romina” and “Sibilla” versus “Silvia” and “Cristina” cultivars using both datasets.The data fusion approach showed improved classification results, highlighting the importance of certain wavenumbers associated with OH, CH, and CH_2_ bonds for distinguishing between cultivars.

In this study, we explored the possibility to use NIR spectroscopy and chemometrics as a method for the rapid phenotyping of strawberry fruits speeding up the selection of material for the breeding process. Even if the fused dataset demonstrated enhanced separation among cultivars, we acknowledged the need for a larger dataset to cover more cultivar variability. Nevertheless, the findings underscored the value of data fusion in enhancing classification performance by reducing non-relevant information. Moreover, this study highlighted the advantages of using near-infrared spectroscopy for cost-effective and rapid assessment of taste and nutritional quality, which can be instrumental in streamlining the breeding process, aligning with consumer preferences. As a future research step, we plan to enlarge the dataset size adding more cultivars, since the outcomes of this study are promising for advancing quality control applications and cultivar classification in the food industry.

## Figures and Tables

**Figure 1 foods-12-03253-f001:**
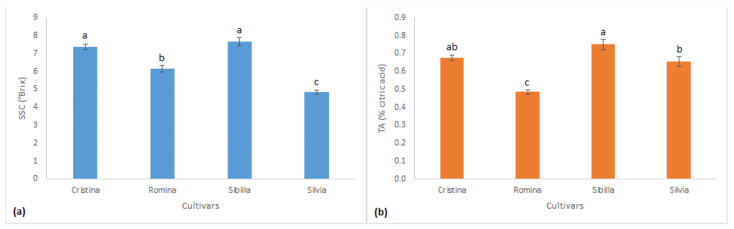
(**a**) Results of fruit soluble solids content (SSC) and (**b**) titratable acidity (TA) for the 4 studied cultivars. Data are expressed as means ± standard error. Different lowercase letters indicate significant differences for *p* < 0.05 (Tukey test).

**Figure 2 foods-12-03253-f002:**
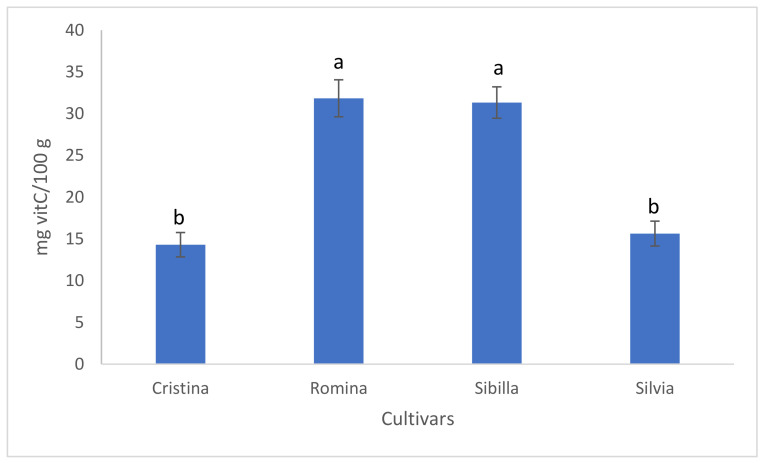
Results of fruit vitamin C content for the 4 studied cultivars. Data are expressed as means ± standard error. Different lowercase letters indicate significant differences at *p* < 0.05 (Tukey test).

**Figure 3 foods-12-03253-f003:**
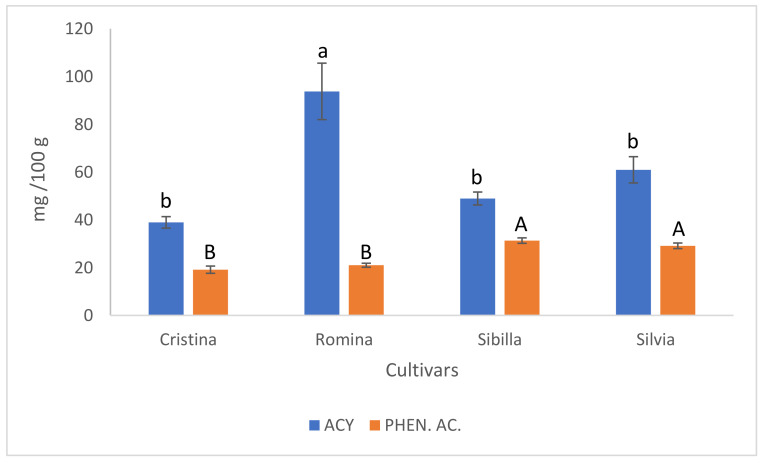
Results of fruit anthocyanins (ACY) and phenolic acids (PHEN. AC.) content for the four studied cultivars. Data are expressed as means ± standard error. Different lowercase letters indicate significant differences for ACY at *p* < 0.05 (Tukey test). Different uppercase letters indicate significant differences for PHEN. AC. at *p* < 0.05 (Tukey test).

**Figure 4 foods-12-03253-f004:**
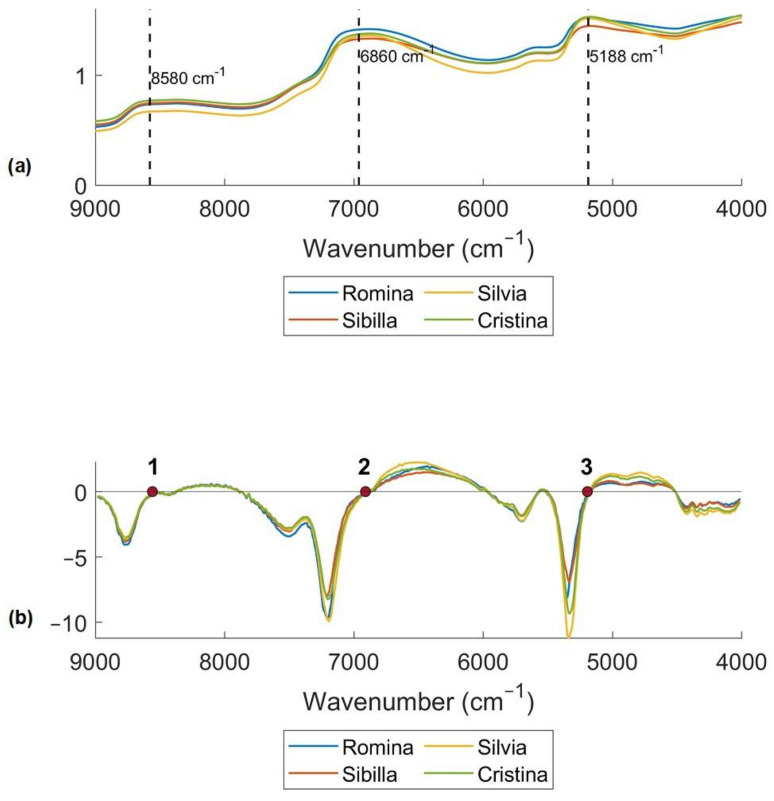
(**a**) Average raw spectra of the four cultivars with important wavenumbers marked with dotted lines and (**b**) average pretreated spectra of the four cultivars using first derivative (Savitzky–Golay filter, 21 smoothing points, 2nd polynomial order). The most important wavenumbers are marked with red dots.

**Figure 5 foods-12-03253-f005:**
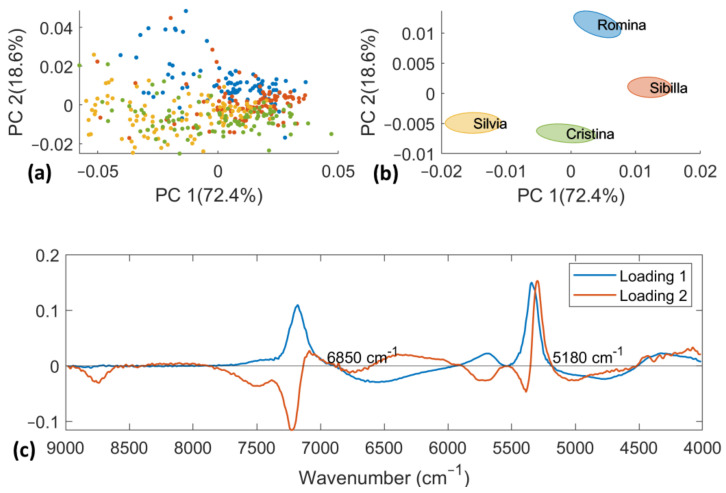
PCA score plot of PC1 vs. PC2 computed on D-NIR dataset (**a**) with standard error ellipses for each cultivar (**b**). PCA loading plot of the two first PCs (**c**).

**Figure 6 foods-12-03253-f006:**
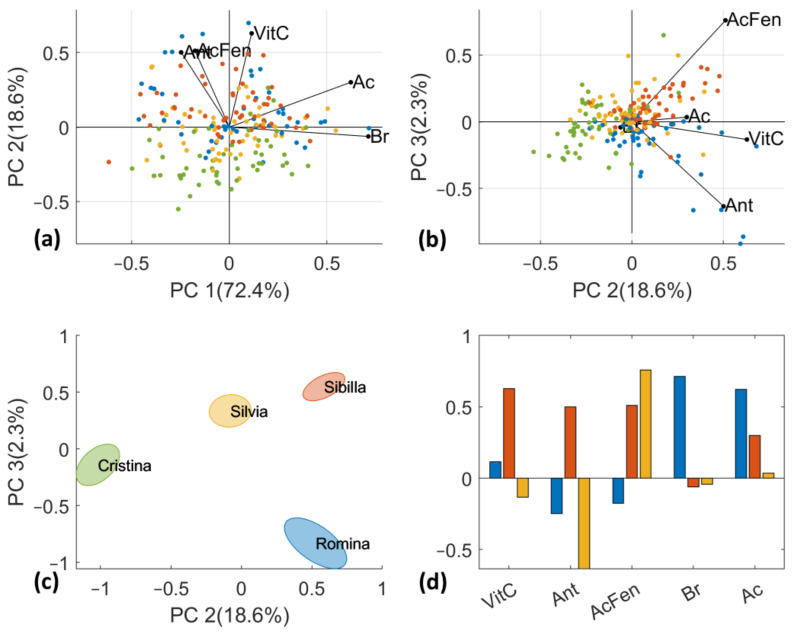
PCA score plots of PC1 vs. PC2 (**a**) and PC2 vs. PC3 (**b**) computed on D-Lab dataset. (**c**) PCA score plot of PC2 vs. PC3 with standard error ellipses for each cultivar. (**d**) PCA loading plot of the three first PCs.

**Figure 7 foods-12-03253-f007:**
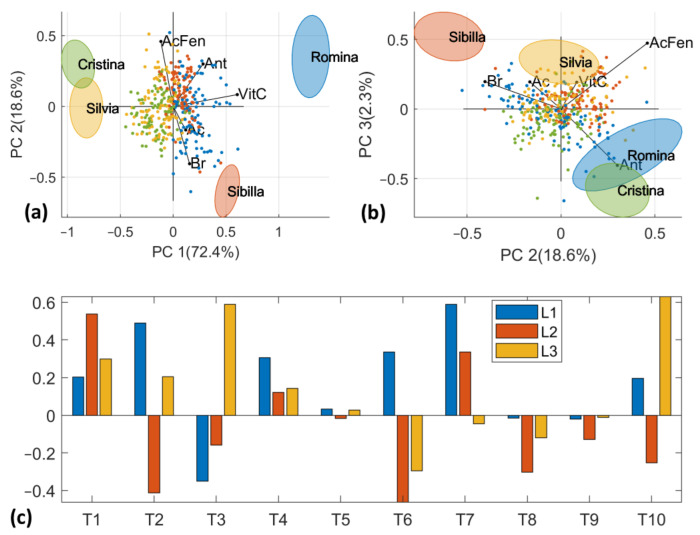
(**a**) PCA score plot of PC1 vs. PC2 computed on D-fused dataset with standard error ellipses for each cultivar and D-Lab loading reconstructed. (**b**) PCA score plot of PC2 vs. PC3 on D-fused dataset with standard error ellipses for each cultivar and D-Lab loading reconstructed. (**c**) PCA loading plot of the three first PCs.

**Figure 8 foods-12-03253-f008:**
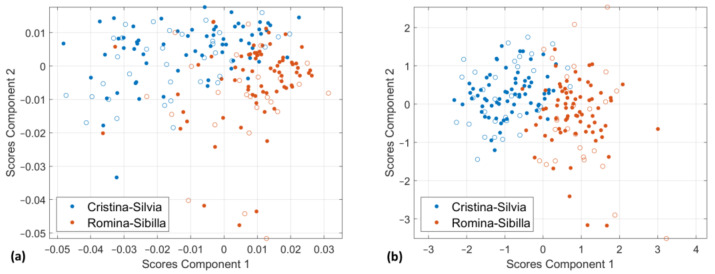
Scores for the first two latent variables of the PLS-DA models obtained by using the spectral dataset (**a**) and D-fused dataset (**b**) for the classification of strawberries samples of Silvia or Cristina cultivars from Sibilla or Romina cultivars (MODE 1). Test samples are represented by empty symbols.

**Figure 9 foods-12-03253-f009:**
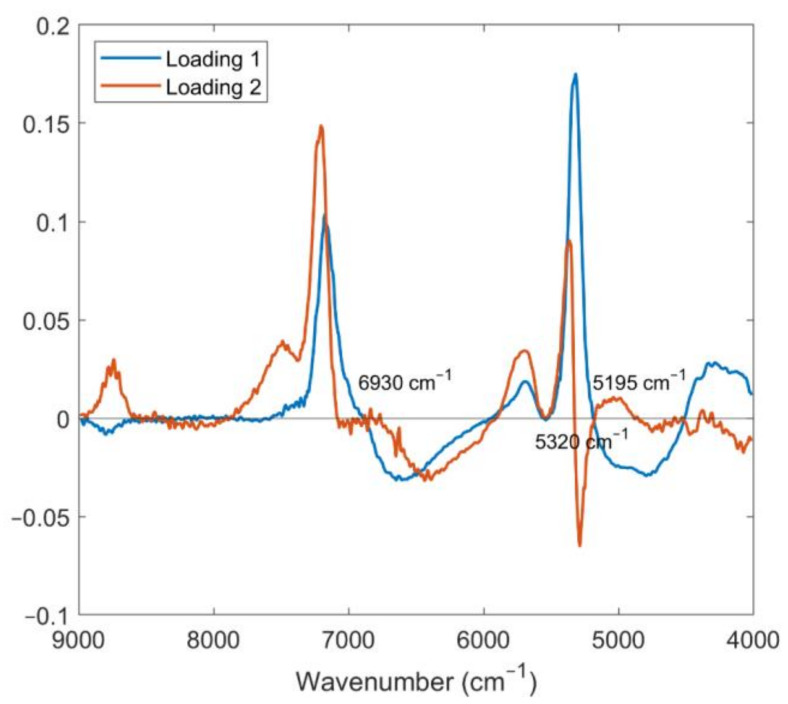
Loading plot of the two first latent variables of the PLS-DA models obtained by using D-NIR dataset. The most relevant wavenumbers for the classification of the strawberry fruits in the two classes are marked in the plot.

**Table 1 foods-12-03253-t001:** Classification results obtained by NIR spectral dataset and D-fused dataset. LVs: latent variables; TNR: true negative rate; TPR: true positive rate.

Model	Cross-Validation				Validation			
9der1	LVs	TNR	TPR	Error	LVs	TNR	TPR	Error
MODE 1	5	93.1	95.4	5.8	5	100.0	100.0	0.0
MODE 2	2	3.1	100.0	22.6	2	89.5	65.9	26.7
MODE 3	2	0.0	100.0	24.1	2	75.0	70.5	28.3
**D-fused**								
MODE 1	3	97.1	94.0	4.4	3	88.9	97.0	6.7
MODE 2	2	17.6	98.1	21.9	2	82.4	88.4	13.3
MODE 3	2	33.3	100.0	16.1	2	87.5	88.6	11.7

## Data Availability

The data presented in this study are available on request from the corresponding author. The data are not publicly available due to ongoing funding projects which not provide public data sharing before the end of the project.

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
