# Peer review of "Application of Near Infrared Spectroscopy for the Rapid Assessment of Nutritional Quality of Different Strawberry Cultivars"

_foods, 2023, doi:10.3390/foods12173253_

Round 1

Reviewer 1 Report

1. Introduction should be revised. Please add previous work on the use of NIR spectroscopy for the evaluation of strawberry quality from the same authors. What is the main difference of the present study with the previous one? The spectral data is coming from the same fruits?

2. Please add the limitation of NIR spectroscopy (especially long wave NIR), especially for measuring fruits with high water content (fresh fruit). The use of frozen fruit is a strategy to solve the problem? I guess the authors utilized the long-wave NIR for spectral data measurement (1000-2500 nm). In this region, the influence of water absorbance is very high. Please mention this in detail in the manuscript.

3. I think it is also important to enhance the strong novelty of the work. The authors may introduce the use of the fusion data technique in this study in the Introduction part.

4. The authors missed describing the NIR spectroscopy acquisition system in the Materials and Methods section. Please add in detail. How the spectral data are obtained? What devices? short or long wave region? grating or FT-NIR? 

5. It is also important to analyze the original spectral data of frozen strawberries. From this point, then explain we need to apply several spectral preprocessing. 

6. How do the authors classify the samples? what is the threshold value for this? 0.5 or using some Bayesian algorithm? Please add this to the manuscript.

7. I did not notice any result of ROC. The authors mentioned ROC in the Materials and Methods. Please clarify.

Author Response

1. Introduction should be revised. Please add previous work on the use of NIR spectroscopy for the evaluation of strawberry quality from the same authors. What is the main difference of the present study with the previous one? The spectral data is coming from the same fruits?

The previous work on the use of NIR spectroscopy for the evaluation of strawberry quality from the same authors were already cited in the previous version of the manuscript (References 15 and 19). However, we revised the introduction to talk a little deeper about these two studies. Each of the three studies deals with a different main topic: in Mancini et al. 2020, the NIR spectroscopy has been applied in same fruits analyzed for SSC, titratable acidity, Firmness and color to evaluate the possibility of building up a good PLS model for the prediction of fruit quality. This result was obtained only for firmness and SSC, while the models obtained for the prediction of color and titratable acidity were not acceptable for screening control. In Mancini et al. 2023, strawberry fruits were analyzed by NIR spectroscopy and were also infected (or not) with B. cinerea spores, and also the SSC content of fruits was measured. In this case, the study demonstrated that there was a high correlation between SSC and B. cinerea susceptibility. Furthermore, it was confirmed that the SSC content can be predicted by PLS regression models. Some of the spectral data used in our current study were the same of spectral data from that study, but not at all. In the current study we applied a completely different strategy to evaluate the possibility of predicting other fruit quality parameters (acidity, vitamin C, anthocyanin and phenolic acid) and SSC as confirmation of the previous studies, by analyzing fruits by NIR and applying the data fusion approach for the data classification.

2. Please add the limitation of NIR spectroscopy (especially long wave NIR), especially for measuring fruits with high water content (fresh fruit). The use of frozen fruit is a strategy to solve the problem? I guess the authors utilized the long-wave NIR for spectral data measurement (1000-2500 nm). In this region, the influence of water absorbance is very high. Please mention this in detail in the manuscript.

The fruit samples were analysed fresh and not frozen (line 143 and line 249) in the spectral range from 1000 to 2500 nm.

A stated in Walsh et al. (2020) (https://doi.org/10.1016/j.postharvbio.2020.111246), the concept of quantitative determination of analyte level in solid materials using near infrared spectroscopy was established by Karl Norris of the United States Department of Agriculture (USDA) in the 1960s, based on work with low moisture agricultural products (e.g. water in grains). USDA work continued into application to high moisture content intact fruit and vegetables, with Birth et al. (1985) publishing on the assessment of the dry matter content on onions, and the group continuing into assessment of total soluble solids of melons (Dull et al., 1989). Until now there have been several studies reporting the use of NIR spectroscopy for the assessment of qualitative and nutritional parameters of vegetables and fruits, with some review reported here: DOI 10.1007/s11694-009-9079-z, DOI 10.3389/fnut.2022.973457). Bibliography research demonstrated the possibility of analyzing fruits despite the high moisture content thanks to the use of chemometrics that allows the extraction of relevant data out of the spectra. This information was added at lines 379-386.

As a consequence, even if hydrated objects have complex hydrogen bonding interactions (e.g. between water, sugar, protein, etc.), in previous studies authors manage in summarizing the band assignments of the major water and sugar (O–H and C–H) vibrations (Magwaza, 2012, Golic 2002).

3. I think it is also important to enhance the strong novelty of the work. The authors may introduce the use of the fusion data technique in this study in the Introduction part.

The strong novelty of the work has been underlined in the last part of the Introduction chapter. This work is aiming at correlating the nutritional characteristics of the strawberry fruits with the NIR spectrometry analysis, and trying to apply the data fusion methodology for the management of this amount of data and a better classification of the results. The great novelty of this work is related to the application of NIR as a non-destructive tool for the evaluation of strawberry nutritional quality, and to the contemporary application of data fusion techniques for the classification of the results.

4. The authors missed describing the NIR spectroscopy acquisition system in the Materials and Methods section. Please add in detail. How the spectral data are obtained? What devices? short or long wave region? grating or FT-NIR? 

A new section (2.9 NIR analysis) has been added to the manuscript.

5. It is also important to analyze the original spectral data of frozen strawberries. From this point, then explain we need to apply several spectral preprocessing. 

A new section ‘3.2. Spectra’ has been added to the manuscript for analyzing both the raw and pretreated spectra.

6. How do the authors classify the samples? what is the threshold value for this? 0.5 or using some Bayesian algorithm? Please add this to the manuscript.

The threshold value used for classifying the samples has been decided using the receiver operating characteristic (ROC) curve. The ROC curve is the plot of the true positive rate (TPR) against the false positive rate (FPR), at various threshold settings, so it is a performance measurement for classification problems at various threshold settings. In fact, for each class of a classifier, threshold values across the interval are applied to outputs and for each threshold, two values are calculated, the True Positive Ratio (or sensitivity), and the False Positive Ratio. Basically, it illustrates the diagnostic ability of a binary classifier system as its discrimination threshold is varied and it selects the one optimizing the classification model. The information has been stated in lines 317-325.

7. I did not notice any result of ROC. The authors mentioned ROC in the Materials and Methods. Please clarify.

Please see the previous answer. ROC curve has been used to assess a threshold limit and, consequently, classification parameters as sensitivity, specificity, accuracy and misclassification rate have been values are computed.

Reviewer 2 Report

The manuscript explores the use of near infrared spectroscopy as a non-destructive method for evaluating the nutritional properties of strawberries. The study analyzes 216 strawberry fruits from four cultivars and develops PLS-DA models for classifying the fruits based on their quality and nutritional characteristics. The research highlights the potential of NIR spectroscopy for phenotyping and monitoring fruit quality in breeding programs. However, manuscript should be revised because of the following concerns:
1. In the introduction, the second paragraph emphasizes the use of near infrared (NIR) spectroscopy as a non-destructive method. While this is a valuable aspect of NIR spectroscopy, it may not be the most significant highlight for this particular research. Considering that the strawberry samples used in this study are not particularly valuable or rare, the emphasis should be placed on the advantages of rapid and convenient testing over conventional methods.
2. Furthermore, it would be beneficial to provide more details about the NIR spectral measurement. It would be helpful to mention the type of instrument used and the specific running parameters employed during the data collection process. These details would enhance the reproducibility of the study and allow readers to better understand the methodology.
3. Regarding the concept of data fusion, it is indeed an intriguing approach. However, it is worth noting that the clustering of the four strawberry cultivars in the fused data is similar to the results obtained from individual D-LAB PCA and D-NIR PCA analyses. This suggests that the fusion technique does not provide additional discriminatory power beyond what can be achieved with the individual datasets. Instead, an interesting avenue for further exploration could be the application of Partial Least Squares Regression (PLSR), particularly using the Partial Least Squares 2 (PLS2) algorithm, which can handle multiple response variables. In this case, utilizing all the nutritional parameters as response variables could yield valuable insights and could be a valuable addition to the manuscript.
4. The authors use 3 different models for dealing with multiclass problem. I suggest use multiclass PLS-DA models since they provide a more comprehensive analysis. When dealing with a dataset that contains multiple classes, comparing the performance of two-class PLS-DA models and multiclass PLS-DA models is an important consideration. Multiclass PLS-DA models, designed to handle datasets with more than two classes, extend the two-class approach to accommodate multiple classes simultaneously. Multiclass PLS-DA models can provide a comprehensive overview of the relationships and differences among all the classes in the dataset. They allow for the identification of discriminant features and patterns across multiple classes, providing a more holistic understanding of the data. Multiclass PLS-DA models are particularly valuable when the objective is to classify samples into multiple classes and explore the interclass relationships.
5. at Line 494, what is rpls stand for?

Author Response

The manuscript explores the use of near infrared spectroscopy as a non-destructive method for evaluating the nutritional properties of strawberries. The study analyzes 216 strawberry fruits from four cultivars and develops PLS-DA models for classifying the fruits based on their quality and nutritional characteristics. The research highlights the potential of NIR spectroscopy for phenotyping and monitoring fruit quality in breeding programs. However, manuscript should be revised because of the following concerns:

  1. In the introduction, the second paragraph emphasizes the use of near infrared (NIR) spectroscopy as a non-destructive method. While this is a valuable aspect of NIR spectroscopy, it may not be the most significant highlight for this particular research. Considering that the strawberry samples used in this study are not particularly valuable or rare, the emphasis should be placed on the advantages of rapid and convenient testing over conventional methods.

We include a paragraph in the Introduction emphasizing the many advantages of non-destructive methods over conventional methods (Lines 54-72).

  1. Furthermore, it would be beneficial to provide more details about the NIR spectral measurement. It would be helpful to mention the type of instrument used and the specific running parameters employed during the data collection process. These details would enhance the reproducibility of the study and allow readers to better understand the methodology.

As also requested by the first reviewer, a new section (2.9 NIR analysis) has been added to the manuscript.

  1. Regarding the concept of data fusion, it is indeed an intriguing approach. However, it is worth noting that the clustering of the four strawberry cultivars in the fused data is similar to the results obtained from individual D-LAB PCA and D-NIR PCA analyses. This suggests that the fusion technique does not provide additional discriminatory power beyond what can be achieved with the individual datasets. Instead, an interesting avenue for further exploration could be the application of Partial Least Squares Regression (PLSR), particularly using the Partial Least Squares 2 (PLS2) algorithm, which can handle multiple response variables. In this case, utilizing all the nutritional parameters as response variables could yield valuable insights and could be a valuable addition to the manuscript.

We thank the reviewer for the suggestion and we may consider applying PLS2 in other studies. Regression is not the aim of this study because we would rather classify the strawberries according to quality and nutritional parameters, but we are not interested in their exact value. In addition as also stated by other authors (e.g. https://doi.org/10.1016/j.arabjc.2014.02.006) PLS1 calculates a separate set of scores and loading vectors for each parameter of interest, so scores and loading vectors are specifically tuned for each parameter, giving more accurate predictions. Considering the low concentration of some parameters (e.g. vitamin C, anthocyanin) they are not easily detected by NIR spectroscopy and regression model could fail.

For all these reasons we preferred to move into a classification approach rather than regression.

  1. The authors use 3 different models for dealing with multiclass problem. I suggest use multiclass PLS-DA models since they provide a more comprehensive analysis. When dealing with a dataset that contains multiple classes, comparing the performance of two-class PLS-DA models and multiclass PLS-DA models is an important consideration. Multiclass PLS-DA models, designed to handle datasets with more than two classes, extend the two-class approach to accommodate multiple classes simultaneously. Multiclass PLS-DA models can provide a comprehensive overview of the relationships and differences among all the classes in the dataset. They allow for the identification of discriminant features and patterns across multiple classes, providing a more holistic understanding of the data. Multiclass PLS-DA models are particularly valuable when the objective is to classify samples into multiple classes and explore the interclass relationships.

Multi-class PLS-DA has been computed without obtaining any good results. Probably it is not possible to have only one model for all classes simultaneously because of the similarities of some cultivars. As also stated in the manuscript, Romina and Sibilla cultivars have similar spectral characteristics making difficult to develop a classification model able to recognize them. For this reason, we have decided to separate the cultivars according to their similarity as the classification model will be in any case used as phenotyping tool in breeding process where it is helpful to have rapid information about qualitative and nutritional characteristics rather than recognize a cultivar from another.

The information was added to the manuscript (lines 510-517).

  1. at Line 494, what is rpls stand for?

The acronym has been stated in the manuscript.

Round 2

Reviewer 1 Report

The revision is acceptable.

Author Response

The authors thank the reviewer for the positive feedback.

Reviewer 2 Report

The authors made extensive changes in this revision and the overall quality improved. However, some issues still need to be clarified further:
1. Line 237, Is "integrative sphere-equipped" refers to the commerically-available Smart NIR Integrating Sphere accessory module from Thermo? if it is, it should be written in a formal and more clear way. 2. Line 243. The 8 cm-1 resolution in 10000 to 4000 cm-1 region resulted 750 data points, which does not match with 1557 absorbance data. Explain the difference. 3. Line 512. A paper must be objective, to be exact, credibility and trustworthiness should be emphasized. "obtaining any good results" is subjective. We should eliminate such personal opinions, emotions, or external influences. I think the subsequent discussions of the aim of the model being developed is justified, but the said multiclass PLS-DA model should be revised with detailed results along with data, and additional, detailed, and unbaised discussions that away from personal beliefs should be added.

Author Response

The authors made extensive changes in this revision and the overall quality improved. However, some issues still need to be clarified further:

  1. Line 237, Is "integrative sphere-equipped" refers to the commerically-available Smart NIR Integrating Sphere accessory module from Thermo? if it is, it should be written in a formal and more clear way.

The sentence has been rephrased.

  1. Line 243. The 8 cm-1 resolution in 10000 to 4000 cm-1 region resulted 750 data points, which does not match with 1557 absorbance data. Explain the difference.

We thank the reviewer for the revision. The typo has now been corrected.

  1. Line 512. A paper must be objective, to be exact, credibility and trustworthiness should be emphasized. "obtaining any good results" is subjective. We should eliminate such personal opinions, emotions, or external influences. I think the subsequent discussions of the aim of the model being developed is justified, but the said multiclass PLS-DA model should be revised with detailed results along with data, and additional, detailed, and unbaised discussions that away from personal beliefs should be added.

The results of the multiclass PLS-DA model have been added to the manuscript (lines 513-515).